# ARDS Mortality Prediction Model Using Evolving Clinical Data and Chest Radiograph Analysis

**DOI:** 10.3390/biomedicines12020439

**Published:** 2024-02-16

**Authors:** Ana Cysneiros, Tiago Galvão, Nuno Domingues, Pedro Jorge, Luis Bento, Ignacio Martin-Loeches

**Affiliations:** 1Nova Medical School, Universidade de Lisboa, 1649-004 Lisbon, Portugal; luis.bento@chlc.min-saude.pt; 2Unidade de Urgência Médica, Hospital de São José, Centro Hospitalar Universitário Lisboa Central, 1169-050 Lisbon, Portugal; 3Instituto Politécnico de Lisboa/Instituto Superior de Engenharia de Lisboa, 1959-007 Lisbon, Portugal; tiagogalvao97@gmail.com (T.G.); nuno.ndomingues@isel.pt (N.D.); pedro.mendes.jorge@isel.pt (P.J.); 4Trinity College Dublin, D02 PN40 Dublin, Ireland; drmartinloeches@gmail.com

**Keywords:** ARDS, imaging, machine learning, deep learning

## Abstract

Introduction: Within primary ARDS, SARS-CoV-2-associated ARDS (C-ARDS) emerged in late 2019, reaching its peak during the subsequent two years. Recent efforts in ARDS research have concentrated on phenotyping this heterogeneous syndrome to enhance comprehension of its pathophysiology. Methods and Results: A retrospective study was conducted on C-ARDS patients from April 2020 to February 2021, encompassing 110 participants with a mean age of 63.2 ± 11.92 (26–83 years). Of these, 61.2% (68) were male, and 25% (17) experienced severe ARDS, resulting in a mortality rate of 47.3% (52). Ventilation settings, arterial blood gases, and chest X-ray (CXR) were evaluated on the first day of invasive mechanical ventilation and between days two and three. CXR images were scrutinized using a convolutional neural network (CNN). A binary logistic regression model for predicting C-ARDS mortality was developed based on the most influential variables: age, PaO_2_/FiO_2_ ratio (P/F) on days one and three, CNN-extracted CXR features, and age. Initial performance assessment on test data (23 patients out of the 110) revealed an area under the receiver operating characteristic (ROC) curve of 0.862 with a 95% confidence interval (0.654–0.969). Conclusion: Integrating data available in all intensive care units enables the prediction of C-ARDS mortality by utilizing evolving P/F ratios and CXR. This approach can assist in tailoring treatment plans and initiating early discussions to escalate care and extracorporeal life support. Machine learning algorithms for imaging classification can uncover otherwise inaccessible patterns, potentially evolving into another form of ARDS phenotyping. The combined features of these algorithms and clinical variables demonstrate superior performance compared to either element alone.

## 1. Introduction

Acute Respiratory Distress Syndrome (ARDS) is a critical medical condition characterized by severe respiratory failure posing a substantial threat with high mortality rates. It can manifest in response to diverse underlying causes, including pneumonia, sepsis, trauma, or inhalation injuries [1]. ARDS is a significant concern in critical care medicine due to its potential for rapid progression and severe respiratory compromise. The Berlin criteria define ARDS by acute lung insult, bilateral chest infiltrates, and hypoxemia not fully explained by other factors [1,2].

While widely accepted, the Berlin criteria need to capture ARDS’s multifaceted nature fully. This syndrome encompasses a spectrum of clinical disorders, diverse physiological abnormalities, varied radiographic presentations, multiple potential microbiological causes, and dynamic evolution over time [3,4]. ARDS exhibits a continuum from the early development of acute lung injury to meeting specific diagnostic criteria [4].

### Diagnostic Criteria and Radiological Evidence

The Berlin criteria assume the use of arterial blood gases, specifically the PaO_2_/FiO_2_ ratio, and require radiological evidence often obtained through chest X-ray (CXR) or chest CT scans. Recently published ESICM definitions have sparked debate and alternative approaches in the medical community, challenging the reliance on chest radiography [5,6].

A prior randomized controlled trial (RCT) revealed no improvement in CXR interpretation following standardized training specific to ARDS. Alternative approaches have been debated, such as accepting unilateral opacities mandating computed tomography (CT) scans and incorporating lung ultrasound in ARDS [6].

Despite being a prevalent cause of acute respiratory failure with high morbidity and mortality, ARDS lacks proven therapeutic options beyond lung protective ventilation [7].

The underlying heterogeneity has prompted research into ARDS sub-phenotypes that may respond to specific treatments [8,9,10].

Recent latent class analysis (LCA) has identified sub-phenotypes with distinct clinical and biological features, emphasizing the importance of precision medicine in addressing ARDS heterogeneity [10].

The lack of a definitive diagnostic test contributes to the broad definition of ARDS, exemplified by histopathological findings in only 45% of autopsied lungs from ARDS patients [11,12].

The debate over steroids in ARDS underscores the need for phenotyping, as specific subtypes, like COVID-19-associated ARDS (C-ARDS), may benefit from steroids [13], whereas plenty of RCTs have not been able to demonstrate clear benefit from steroids in ARDS [14,15].

Prediction of ARDS remains challenging, and the heterogeneity of its etiology often hampers clinical trials. Machine learning, especially in radiology, is gaining prominence and has been employed in ARDS diagnosis [16,17,18,19] as well as studying recruitability and response to PEEP [17,20].

Our study aims to analyze evolving ventilation settings, PaO_2_/FiO_2_, and chest radiography in patients with C-ARDS. The hypothesis is that the combination of chest X-rays with ventilation settings and arterial blood gases can effectively predict ARDS mortality, contributing to more comprehensive understanding of this complex syndrome.

## 2. Materials and Methods

### 2.1. Study Design

A retrospective cohort study was conducted on mechanically ventilated C-ARDS patients admitted to the intensive care units of Hospital São José and Hospital Curry Cabral between April 2020 and January 2021. This is a university hospital trust with 217 beds and 55 ICU beds. Patient data were acquired through electronic medical records.

### 2.2. Inclusion and Exclusion Criteria

Patients on extracorporeal membrane oxygenation (ECMO) were excluded due to potential differences in ventilation data arising from lung rest ventilation strategies. The analysis focused on the first 72 h of ICU admission when SARSCOV was the sole infectious isolate. Patients diagnosed with ARDS according to the Berlin definition requiring invasive mechanical ventilation for at least 48 h were included. Exclusion criteria encompassed age below 18, pregnancy, any other contributing causes of ARDS, synchronous respiratory infection with other agents, death within the initial 48 h of ICU admission, and lack of data on ventilation settings or poor-quality chest radiographs.

### 2.3. Primary Outcome

The primary outcome was defined as all-cause mortality.

### 2.4. Data Acquisition

Patient data included age, gender, arterial blood gases, and ventilation settings during the first day (PS d1) and between 48 and 72 h (PS d3) of invasive mechanical ventilation. Portable chest radiographs (CXRs) at day one (CXR d1) and day three (CXR d3) were processed using Gaussian blur filtering and contrast-limited adaptive histogram equalization. These techniques aim to achieve noise reduction and image contrast optimization in the regions of interest and improve accuracy in image classification tasks using neural networks [21].

The lung area was segmented and properly resized, and normalized CXR d1 and CXR d3 were concatenated. A pre-trained DenseNet121 convolutional neural network (CheXNet) was employed for transfer learning, providing reliable analysis and detection of 14 thoracic-related pathologies. Deep learning features (DLFs) were extracted and coupled with clinical variables (CVs) to construct two machine learning models: logistic regression (LogReg) and a multilayer perceptron (MLP). Age, PaO_2_/FiO_2_ ratio on the third day (P/F d3) of invasive mechanical ventilation, and deep learning features (DLFs) were used in the final models.

### 2.5. Sample Split and Imputation

Sample data were randomly split into training (85 patients) and test (23 patients) groups. Missing data were imputed by the median after splitting. Imputation was individually applied to each subset of cross-validation to reduce risk of bias and variance. Other methods were not used because the dataset was relatively small, with the risk of outliers.

### 2.6. Statistical Analysis

Models were developed using logistic regression (LogReg), multilayer perceptron (MLP), support vector machine (SVM), and random forest (RndForest) algorithms for comparison. Sequential feature selection, employing logistic regression with five-fold cross-validation in the training group, was performed to avoid model overfitting. For comparison purposes, a model was created using only clinical data with age and P/F d3 (Model B).

## 3. Results

A total of 110 patients were enrolled in the study, with a mean age of 63.2 ± 11.92 (26–83 years). The gender distribution revealed an expected predominance of males, constituting 61.2% (*n* = 68) of the cohort. Severe ARDS was present in 25% (17) of patients. The overall mortality rate was 47.3% (52). Further epidemiological characteristics are displayed in Figure 1 and Table 1.

Figure 1 shows the majority of patients had moderate ARDS based on their P/F ratio, with severe ARDS more common in the 60–80 age group and more common in males.

Table 1 shows 61.2% of patients were male and overall mortality was 47.3%. The average age was 63.2% (min 26; max 83) and the average P/F ratio on admission was 148.5 (min 51 max 297). The average P/F ratio increased on day 3 (163.73). The average PEEP on day 1 was 12.76 and was 11.61 on day 3.

Table 2 shows how DLFs increased sensitivity from 0.643 (model B) to 0.714 (model A); specificity from 0.711 to 0.778; the positive predictive value (PPV) from 0.675 to 0.75 and the AUC from 0.701 to 0.77. Regarding the Bayesian *t*-test results, the probability of model A being better than model B was 0.739.

### 3.1. Logistic Regression Analysis

Utilizing logistic regression analysis, the most influential variables were identified as age, PaO_2_/FiO_2_ at day one (P/F d1), PaO_2_/FiO_2_ at day three (P/F d3), and deep learning image features (DLFs). Cross-validation using P/F d3, age, and DLFs demonstrated optimal performance metrics.

### 3.2. Model Performance

The models incorporating DLFs exhibited an 89% probability of superior accuracy for logistic regression (LogReg) and 82% for multilayer perceptron (MLP) compared to the clinical variables-only model (Model B). Within the internal test group (23 patients), the LogReg model emerged as the most robust, yielding an area under the ROC curve (AUC) of 0.862 with a 95% confidence interval [0.654, 0.969], an accuracy of 0.783 (95% CI [0.563, 0.926]), and an F1 score of 0.783 (95% CI [0.563, 0.926]). Detailed results are shown in Table 3.

### 3.3. Comparative Analysis

The same analysis was repeated using only clinical variables and P/F d3, revealing an AUC of 0.77. Notably, enhanced classification performance was observed when incorporating CNN-extracted image features.

### 3.4. Cross-Validation Significance

Cross-validation, which is crucial for reducing bias and accounting for outliers within the same population, was employed. This approach is particularly significant in small-population studies, ensuring robustness in the analysis and interpretation of the results, as shown in Figure 2 and Table 4.

Figure 2 shows the cross-validation sensitivity and specificity of both models by logistic regression and MLP, with model A, which includes DLF, performing better than model B.

Table 4 uses cross-validation, which is important when using small populations. Again, the contribution of DLF is sustained, with AUC improving from 0.701 (CI 0.593–0.794) to 0.77 (CI 0.667–0.853). Improvement was greater in PPV, which rose to 0.75 (CI 0.558–0.839) from 0.675 (CI 0.509–0.814), and than NPV, which improved from 0.714 (CI 0.554–0.843) to 0.744 (CI 0.596–0.861).

Figure 3 shows the individual contribution of the three selected features to the final model using logistic regression.

Figure 3 shows the contribution of the three selected features to the model using logistic regression. Increasing age and decreasing P/F are therefore associated with mortality (the positive outcome). DLFs result from the concatenated images of CXR at day 1 and day 3.

## 4. Discussion

In the evolving landscape of Acute Respiratory Distress Syndrome (ARDS) research, the utilization of machine learning techniques, particularly in conjunction with chest radiography, holds promise for providing valuable insights into the prediction of mortality and severity in patients with COVID-19-associated ARDS (C-ARDS). This study explored the association between image features extracted from chest radiographs and patient outcomes, emphasizing the evolving nature of chest imaging in the first 72 h of invasive mechanical ventilation. Our main message is that we proved that integrating deep learning image features in the logistic regression model exhibited superior predictive accuracy, providing valuable insights into mortality prediction in C-ARDS patients. The robust performance metrics, especially within the internal test group, underscore the potential clinical utility of the proposed model.

In the dynamic and continually advancing field of ARDS research, integrating machine learning techniques, particularly in tandem with chest radiography, presents a transformative avenue with substantial promise. This synergistic approach has the potential to yield profound insights into predicting not only the mortality outcomes but also the severity of ARDS in afflicted patients. The confluence of machine learning and chest radiography stands out as a cutting-edge paradigm poised to enhance our understanding of ARDS’s intricate dynamics and nuanced manifestations, ultimately contributing to more effective and personalized approaches to patient management.

The evolving landscape of ARDS research reflects a growing recognition of the complexities inherent in this critical medical condition. Traditional methodologies have encountered challenges in addressing the heterogeneity of ARDS, emphasizing the need for innovative and sophisticated techniques to unravel its multifaceted nature. In this context, the amalgamation of machine learning and chest radiography emerges as a revolutionary strategy, offering a comprehensive and nuanced perspective on the predictive factors influencing both mortality and severity in ARDS patients.

Machine learning, with its capacity to discern patterns and relationships within vast datasets, complements the intricacies of ARDS by providing a data-driven framework for analysis. Integrating these advanced computational methods with chest radiography, a widely accessible imaging modality, establishes a powerful synergy. This combined approach capitalizes on the detailed information embedded in radiographic images, enabling the identification of subtle yet clinically significant features that may elude conventional diagnostic and prognostic assessments.

The promise lies not only in the ability to predict mortality outcomes but also in gauging the severity of ARDS, a crucial aspect that influences the trajectory of patient care. By harnessing the potential of machine learning algorithms to analyze intricate radiographic details, the predictive model becomes more adept at discerning the nuances of disease progression, thereby contributing to more nuanced understanding of ARDS severity.

This integrated approach is not merely confined to a technological juxtaposition but represents a fundamental shift in the paradigm of ARDS research. It transcends the conventional boundaries of diagnostic and prognostic methodologies, offering a holistic and data-driven framework that aligns with the evolving complexities of ARDS pathophysiology. As a result, this innovative synthesis of machine learning and chest radiography stands poised to redefine the landscape of ARDS research, ushering in a new era of precision medicine and personalized patient care.

The challenges in ARDS treatment have been underscored by its inherent heterogeneity. The RECOVERY trial [13] demonstrated a mortality benefit of steroids in mechanically ventilated patients fulfilling the Berlin criteria, particularly in those with COVID-19-associated ARDS (C-ARDS), suggesting a potential subgroup homogeneity. However, the applicability of steroids across all ARDS cases remains uncertain, as evidenced by varying outcomes in ARDS secondary to influenza [22,23].

Our study delves into the predictive capacity of deep learning features extracted from chest radiographs, surpassing the predictive capability of the P/F ratio in terms of mortality. This finding aligns with the evolving trend in ARDS research, which emphasizes the importance of integrating lung morphology assessments in patient management. Studies have traditionally focused on radiographic assessment of lung edema (RALE), which was validated using patients in the ARDS Network Fluid and Catheter Treatment Trial [24]. Studies have found the RALE score to be correlated with ARDS severity [25,26] and survival [27], but this score requires specific training to reduce observer variability.

Recent studies using imaging patterns to tailor ventilation strategies have had mixed outcomes. The LIVE trial, for instance, indicated that personalization based on CT-based image classification did not decrease mortality, potentially due to misclassification and subsequent mismatch in ventilator strategies [28]. Automated approaches, as performed in our study, identify regions of interest with less risk of misclassification.

Our study sheds light on the evolving nature of chest imaging within the first 72 h of invasive mechanical ventilation, revealing a strong correlation with mortality. Unlike baseline images, [29,30], this temporal relationship aligns with recent studies highlighting the prognostic value of changes in imaging parameters over time [29].

Lung ultrasound has also emerged as promising in distinguishing between focal and non-focal ARDS [31,32], and trials are ongoing regarding lung ultrasound patterns and personalized mechanical ventilation. However, lung ultrasound is operator-dependent and more time-consuming.

Integrating multi-source data becomes imperative as ARDS research transitions towards a phenotyping strategy. This study, leveraging data readily available in the ICU, demonstrates our potential to achieve a robust predictive model. Nevertheless, the study’s limitations, including its retrospective nature and relatively small population size, warrant prospective validation to enhance its clinical utility; an external validation cohort is underway, which will include ARDS patients from a Dublin ICU. The data assessed will include more outcomes such as ventilator-free days and length of ICU stay.

Given its high volume of images and cost-effectiveness, chest radiography emerges as an appealing modality for machine learning applications. Unlike chest tomography, bedside chest radiographs offer the advantage of being readily accessible and conducive to repeated examinations, facilitating the assessment of disease progression. However, limitations in available data necessitated the selection of clinical variables. Notably, comprehensive data on ventilation parameters, including driving pressure and plateau pressure, were not uniformly obtainable within the specified timeframe. Consequently, the PaO_2_/FiO_2_ ratio (P/F ratio) was chosen, despite its susceptibility to influence from positive end-expiratory pressure (PEEP).

## 5. Conclusions

In conclusion, our findings underscore the significance of integrating chest radiography and machine learning in early mortality prediction in C-ARDS. The evolving aspect of chest imaging, particularly within the first 72 h of invasive mechanical ventilation, emerges as a critical determinant of patient outcomes. As ARDS research advances towards a phenotyping strategy, we anticipate future studies will build upon our findings, combining multi-source data to refine treatment strategies for specific patient subgroups. Prospective validation is essential for establishing the clinical applicability of our predictive model, ultimately enabling early discussions and strategic planning for patients at higher risk of mortality in the critical care setting.

## Figures and Tables

**Figure 1 biomedicines-12-00439-f001:**
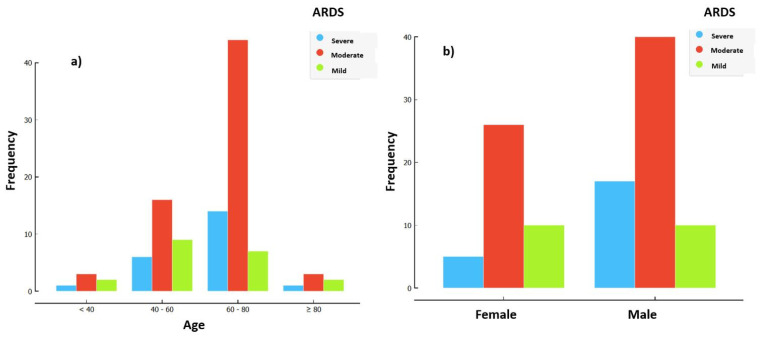
Distribution of ARDS severity at day 1 of IMV based on pf_d1. (**a**) Distribution of ARDS severity by age. (**b**) Distribution of ARDS severity by gender.

**Figure 2 biomedicines-12-00439-f002:**
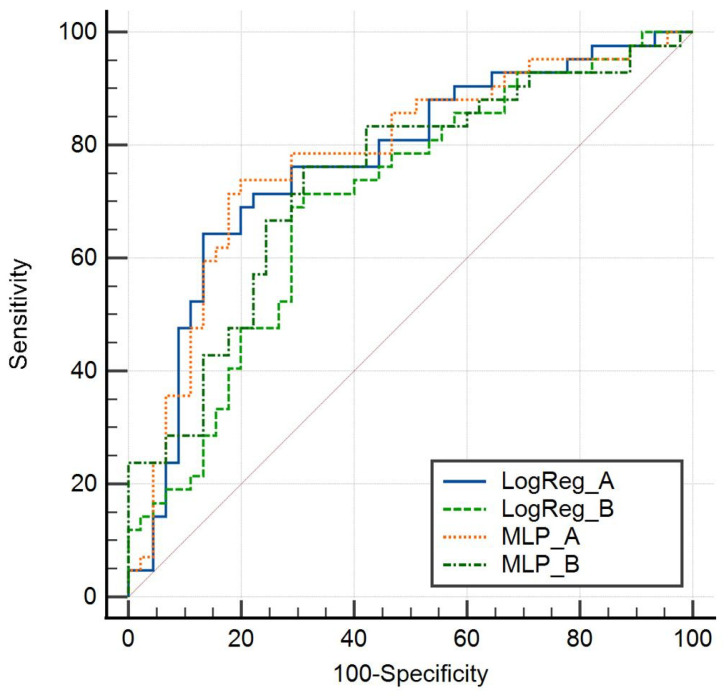
Cross-validation sensitivity and specificity of model A and model B by logistic regression and MLP.

**Figure 3 biomedicines-12-00439-f003:**
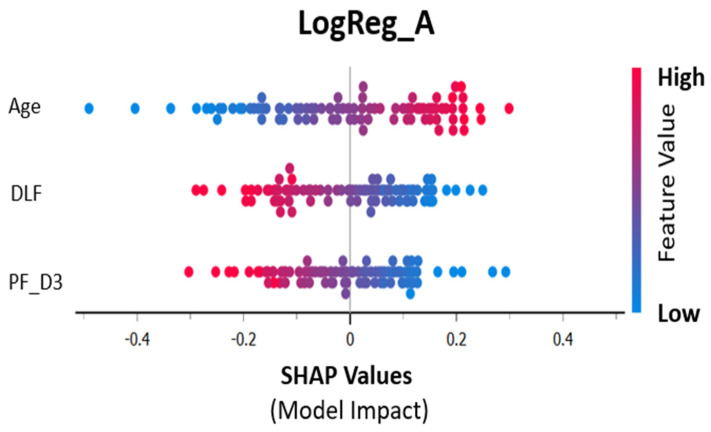
SHAP values (contribution to training prediction).

**Table 1 biomedicines-12-00439-t001:** Descriptive statistics of the total database.

Variables	Categorical (*n* = 110)					
Mortality (target: NS)	**Non-Survivors**		***n* = 52**		**47.3%**	
	Survived		*n* = 58		52.7%	
Gender	Male		*n* = 68		61.2%	
	Female		*n* = 42		38.2%	
Continuous						
	**Average. ± s.d**	**Median**	**Mode**	**Min–Max**	**C_v_**	**Missing (%)**
Age	63.2 ±11.87	64	64	26:83	0.19	0
P/F_d1	148.50± 52.92	147.5	150	51:297	0.36	2 (2%)
P/F_d3	163.73 ± 53.07	160	118	52:365	0.32	1 (1%)
PS_d1	16.22 ± 3.19	16	16	8:26	0.20	3 (3%)
PS_d3	15.49 ± 3.50	18	16	0:26	0.23	7 (6%)
PEEP_d1	12.76 ± 3.29	14	14	6:20	0.26	0 (0%)
PEEP_d3	11.61 ± 3.49	12	12	0:22	0.30	5 (5%)
PaCO2_d1	45.84 ± 11.36	44.6	46	18.1:89	0.25	1 (1%)
PaCO2_d3	47.32 ± 9.61	46.2	40.3	26.4:90.6	0.20	2 (2%)

*n* = number of samples; NS = non-survivor; mean. ± d.p = mean ± standard deviation; min– max = minimum and maximum values of the variable; Cv = coefficient of variation; missing (%) = percentage of missing samples. The target class is shown in bold.

**Table 2 biomedicines-12-00439-t002:** Average of the performance metrics of the LogReg_A and LogReg_B models with their Bayesian *t*-test results.

Model	AUC_méd_	AUC_p_	CA	Ss	PPV	Sp	F1
LogReg_A	0.770	0.784	0.747	0.714	0.750	0.778	0.732
LogReg_B	0.701	0.751	0.678	0.643	0.675	0.711	0.659
Bayesian *t*-test	P_AB	0.739	0.886	0.769	0.875	0.758	0.864
	P_BA	0.261	0.114	0.231	0.122	0.242	0.136
	P_5%__AB	0.364	0.630	0.568	0.734	0.547	0.713
	P_5%__BA	0.062	0.026	0.104	0.204	0.107	0.056
	P_neg5%_	0.574	0.453	0.328	0.230	0.346	0.230

P_AB = probability of model A being better than B; P_BA = probability of model B being better than model A; P_5%__AB = probability of model A being better than model B considering a negligible 5% performance difference; P_5%__BA = probability of model B being better than A considering a negligible 5% performance difference; P_neg5%_ = probability of the models being identical considering a negligible performance difference of 5%. CA = classification accuracy; Ss = sensitivity, Sp = specificity; PPV = positive predictive value; F1 = harmonic mean of PPV and sensitivity.

**Table 3 biomedicines-12-00439-t003:** Performance metrics of the final model LogReg_A in the training group (LogReg_train), in cross-validation (LogReg_CV), and in the test group (LogReg_Test). Δ_CV_Train_ = difference between metrics from the training group ad cross-validation. Δ_Test Train_ = difference between the metrics of the training group and test group.

Metrics	LogReg_Train95%CI	LogReg_CV95%CI	LogReg_Test95%CI	Δ_CV_Train_	Δ_Test Train_
AUC_med_	0.820 [0.723–0.894]	0.770 [0.667–0.853]	0.862 [0.654–0.969]	−0.092	0.042
AUC_p_	0.820	0.784	0.862	−0.036	0.042
CA	0.782 [0.681–0.863]	0.747 [0.643–0.834]	0.783 [0.563–0.926]	−0.035	−0.0
F1	0.776 [0.672–0.859]	0.732 [0.623–0.824]	0.783 [0.563–0.926]	−0.044	0.007
PPV	0.767 [0.623–0889]	0.750 [0.588–0.873]	0.692 [0.385–0.909]	−0.026	0.084
NPV	0.795 [0.647- 0.902]	0.744 [0.596- 0.861]	0.900 [0.555–0.997]	−0.051	0.105
Ss	0.787 [0.633–0.898]	0.714 [0.554–0.843]	0.900 [0.555–0.997]	−0.073	0.113
Sp	0.778 [0.629–0.888]	0.778 [0.629–0.888]	0.692 [0.385–0.909]	0.000	−0.086

positive predictive value (PPV), negative predictive value (NPV), sensitivity (Ss), specificity (Sp), F1 = harmonic mean of PPV, sensitivity, and classification accuracy (CA).

**Table 4 biomedicines-12-00439-t004:** Cross-validation analysis by logistic regression.

Metric	LogReg_A	LogReg_B	Δ_A-B_
AUC_med_	0.770 95%CI [0.667–0.853]	0.701 95%CI [0.593–0.794]	0.069
CA	0.747 95%CI [0.643–0.834]	0.678 95%CI [0.569–0.774]	0.069
PPV	0.750 95%CI [0.588–0.873]	0.675 95%CI [0.509–0.814]	0.075
Sensitivity	0.714 95%CI [0.554–0.843]	0.643 95%CI [0.480–0.784]	0.071
NPV	0.744 95%CI [0.596–0.861]	0.714 95%CI [0.554- 0.843]	0.081
Specificity	0.778 95%CI [0.629–0.888]	0.711 95%CI [0.560–0.834]	0.067
F1	0.732 95%CI [0.623–0.824]	0.659 95%CI [0.546–0.760]	0.073

positive predictive value (PPV); negative predicted value (NPV); sensitivity (Ss); specificity (Sp); classification accuracy (CA); F1 = harmonic mean of PPV and sensitivity.

## Data Availability

The debase is still being used for research and other outcomes but will be shared in the future.

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
