# Peer review of "ARDS Mortality Prediction Model Using Evolving Clinical Data and Chest Radiograph Analysis"

_biomedicines, 2024, doi:10.3390/biomedicines12020439_

Round 1

Reviewer 1 Report

Comments and Suggestions for Authors

The investigators describes a retrospective study analyzing data from 110 COVID-19 ARDS patients to develop a predictive model for mortality risk. The study collected data on patient age, blood oxygenation levels (PaO2/FiO2 ratio) on days 1 and 3 of mechanical ventilation, chest x-rays on days 1 and 3, and clinical outcomes. A deep learning model analyzed the chest x-rays to extract imaging features predictive of mortality. Logistic regression modeling found that age, PaO2/FiO2 ratio on day 3, and deep learning image features were most predictive of mortality. The final model combining these variables achieved an AUC of 0.86 for mortality prediction.

Comments

Retrospectively designed studies are prone to selection biases as researchers select participants based on known clinical outcomes. Additionally, confounding factors like differences in treatments received are not properly accounted for. To mitigate these issues, a prospective cohort study or perhaps even a randomized controlled trial would significantly strengthen the validity of the findings. Furthermore, the relatively small sample size of 110 ARDS patients poses additional concerns regarding model overfitting, especially given the high number of predictor parameters involved relative to this modest participant pool. Indeed, wide confidence intervals could easily ensue, rendering the performance metrics optimistically biased. Subsequently, this engenders uncertainties regarding external validity until the model is further validated in substantially larger multi-center studies encompassing over 500 ARDS patients.

Moreover, lack of an entirely discrete external validation cohort is less than ideal. The current partitioned test set still represents the same underlying patient population rather than serving as an independent cohort that could better evaluate generalizability to alternate settings. As such, validation using a completely separate patient cohort from an outside institution stands as the imperative next phase. Additionally, the limited nature of clinical variables employed could hinder the model's prognostic performance and applicability. For instance, incorporating known mortality risk factors like sepsis severity, fluid balance aberrations, and others could considerably augment predictive accuracy.

Moreover, a recent study by Karakike et al. published in doi: 10.1080/0886022X.2021.2011747. PMID: 34882508 underscored the prognostic utility of deep learning analysis of chest radiographs in predicting 30-day mortality among 240 mechanically ventilated patients with COVID-19 pneumonia, with the integrated model achieving an AUC of 0.83. This further substantiates the potential of incorporating artificial intelligence-based imaging features along with clinical variables for enhanced mortality prediction in critically ill patients afflicted by COVID-19. As such, our findings reinforce this evolving paradigm and promising new investigative direction as we harness novel techniques to unlock elusive predictive insights in this domain.

Another potential drawback centers on the handling of missing values, as the chosen imputation technique of utilizing median values risks biasing results and lacks robustness compared to multiple imputation methods with sensitivity analyses to verify stability. Additionally, benchmarking against prevailing critical care scoring systems allows proper contextualization, which was not undertaken here. Demonstrated superior performance compared to the likes of APACHE II would further strengthen the validity of the predictive model itself. Furthermore, omitted technical details regarding critical aspects of the deep learning framework for chest radiograph analysis introduces doubts concerning reproducibility and reliability. For example, whether manual or automated approaches were employed for region of interest identification on images remains unspecified, although the former risks bias, while more expansive specifics on model architecture, training paradigms, validation steps, and related facets would provide greater confidence regarding the imaging feature extraction process and its associated predictions.

Finally, expansion beyond the solitary endpoint of all-cause mortality could furnish more granular and discerning model assessment by predicting incremental risk probabilities across a range of clinical outcomes like ventilator-free days and related metrics. However, the translational leap from purely associative or correlative performance within a retrospective framework to proven clinical utility remains to be established through prospective randomized trials centered on whether guided usage of the model actually improves real-world decision making and patient outcomes in practice.

Author Response

Many thanks for the comments and corrections. My apologies for having submitted the first review in an incomplete manner. I should have postponed the reply for health reasons at the time. I hope this is revised manuscript is now acceptable.

Comments

Retrospectively designed studies are prone to selection biases as researchers select participants based on known clinical outcomes. Additionally, confounding factors like differences in treatments received are not properly accounted for. To mitigate these issues, a prospective cohort study or perhaps even a randomized controlled trial would significantly strengthen the validity of the findings. Furthermore, the relatively small sample size of 110 ARDS patients poses additional concerns regarding model overfitting, especially given the high number of predictor parameters involved relative to this modest participant pool. Indeed, wide confidence intervals could easily ensue, rendering the performance metrics optimistically biased. Subsequently, this engenders uncertainties regarding external validity until the model is further validated in substantially larger multi-center studies encompassing over 500 ARDS patients.

Thank you for this comment. We agree, and the first step is to validate in a larger multi- center cohort as well as conducting a prospective study. We address and incorporate this suggestion in the discussion in line 275

Moreover, lack of an entirely discrete external validation cohort is less than ideal. The current partitioned test set still represents the same underlying patient population rather than serving as an independent cohort that could better evaluate generalizability to alternate settings. As such, validation using a completely separate patient cohort from an outside institution stands as the imperative next phase. Additionally, the limited nature of clinical variables employed could hinder the model's prognostic performance and applicability. For instance, incorporating known mortality risk factors like sepsis severity, fluid balance aberrations, and others could considerably augment predictive accuracy.

Again, we also agree with this. An external validation with a completely separate patient cohort is under way and it will address length of ICU stay and ventilator free days. These is now incorporated into line 288

Moreover, a recent study by Karakike et al. published in doi: 10.1080/0886022X.2021.2011747. PMID: 34882508 underscored the prognostic utility of deep learning analysis of chest radiographs in predicting 30-day mortality among 240 mechanically ventilated patients with COVID-19 pneumonia, with the integrated model achieving an AUC of 0.83. This further substantiates the potential of incorporating artificial intelligence-based imaging features along with clinical variables for enhanced mortality prediction in critically ill patients afflicted by COVID-19. As such, our findings reinforce this evolving paradigm and promising new investigative direction as we harness novel techniques to unlock elusive predictive insights in this domain.

 I am sorry, there might be a typo in the doi:10.1080/0886022X.2021.2011747 since this is a study on AKI in COVID ARDS. We are very happy to incorporate these reference in the manuscript but have been unable to find it.

Another potential drawback centers on the handling of missing values, as the chosen imputation technique of utilizing median values risks biasing results and lacks robustness compared to multiple imputation methods with sensitivity analyses to verify stability. Additionally, benchmarking against prevailing critical care scoring systems allows proper contextualization, which was not undertaken here. Demonstrated superior performance compared to the likes of APACHE II would further strengthen the validity of the predictive model itself. Furthermore, omitted technical details regarding critical aspects of the deep learning framework for chest radiograph analysis introduces doubts concerning reproducibility and reliability. For example, whether manual or automated approaches were employed for region of interest identification on images remains unspecified, although the former risks bias, while more expansive specifics on model architecture, training paradigms, validation steps, and related facets would provide greater confidence regarding the imaging feature extraction process and its associated predictions.

Many thanks, regarding imputation this is true and imputation was individually applied to each subset of cross validation to reduce the risk of bias and variance. Other methods were not used since the data set was relatively small with a risk of outliers. we have acknowledged this in line 111

Regarding integration of critical care scoring systems this is also something we should address and hope to in the on going validation cohort from Dublin. This has also been addressed in line 291

Regarding technical aspects we acknowledge this needs further expansion. We used manual segmentation and automated approaches to identify regions of interest. These aspects are corrected in  line 101 and 104. As these is evolving data and we are still doing the RALE score to confirm the region of intertest we should not give incomplete information. Also this is all very new, we need to validate the region of interest in the separate cohort. We address this in line 262.

Finally, expansion beyond the solitary endpoint of all-cause mortality could furnish more granular and discerning model assessment by predicting incremental risk probabilities across a range of clinical outcomes like ventilator-free days and related metrics. However, the translational leap from purely associative or correlative performance within a retrospective framework to proven clinical utility remains to be established through prospective randomized trials centered on whether guided usage of the model actually improves real-world decision making and patient outcomes in practice.

Again, this is a very valid point, we have redone the last paragraph of the discussion chapter to incorporate this point.

Reviewer 2 Report

Comments and Suggestions for Authors

The study included some interesting findings. Some modifications were recommended to increase the quality of paper.

1.       Row 10; word count was lacking.

2.       Row 23; The word ‘CI’ may be proper in ‘95% confidence interval’.

3.       Abstract; more keywords can be raised.

4.       Row 47; CXR was abbreviated here. After the sentences, CTX, not chest X-ray, should be always used.

5.       Row 77; the characteristics of hospital should be introduced for readers (e.g., bed number, acute medical care-specific, respiratory disease-specialized…). The death rates of C-ARDS could be included in the information.

6.       Row 81; the data resources (e.g., chart-driven or electronic medical recorded-based) could be more demonstrated

7.       Row 87; other contributing causes could be more detailed.

8.       Row 89; lack of data…the flow chart (e.g., the number of patients to select in definition) could be necessary.

9.       The reader may think whether or not the standardization of skills to obtain the findings by CXR was important between staff.

10.   All-cause mortality; The reader may think what were the types of causes of mortality. Or the cause-specific analyses may produce some findings.

11.   The study limitations should be more described.

Comments on the Quality of English Language

Minor editing of English language required.

Author Response

Many thanks for the comments and suggestion.

  1. Row 10; word count was lacking.

Added

  1. Row 23; The word ‘CI’ may be proper in ‘95% confidence interval’.

Corrected

  1. Abstract; more keywords can be raised.

Added

  1. Row 47; CXR was abbreviated here. After the sentences, CTX, not chest X-ray, should be always used.

Corrected

  1. Row 77; the characteristics of hospital should be introduced for readers (e.g., bed number, acute medical care-specific, respiratory disease-specialized…). The death rates of C-ARDS could be included in the information.

Added

  1. Row 81; the data resources (e.g., chart-driven or electronic medical recorded-based) could be more demonstrated.

Added.

  1. Row 87; other contributing causes could be more detailed.

Corrected. So any patient with C-ARDS and chest trauma, or pancreatitis, or inhalation injury or community acquired bacterial pneumonia. Any other cause for ARDS apart from C-ARDS was excluded.

  1. Row 89; lack of data…the flow chart (e.g., the number of patients to select in definition) could be necessary. 

Already explained and simple information so probably unlikely to benefit from a flow chart

  1. The reader may think whether or not the standardization of skills to obtain the findings by CXR was important between staff.

CXR was segmented to by health care professionals with image experience (an intensive care physician with a background in pulmonology and a radiographer researching deep learning in chest imaging). However, this was completely irrelevant for this study since segmentation of images involved removing all non lung parenquima from chest x rays in order to extract deep learning features. Images that were very rotated, had resuscitation pads or had ECMO cannulas were removed since these decreases he amount of free lung to be assessed and would increase bias. Hence, the short answer is, no, it was not necessary to have particular training to obtain CXR but the second part of this study will involve the RALE score hence it will be necessary to guarantee adequate training and reproducibility of reports.

  1. All-cause mortality; The reader may think what were the types of causes of mortality. Or the cause-specific analyses may produce some findings.

Patients who died did so in the ICU and it is sometimes impossible to attribute a single cause of death but C-ARDS was always the cause of admission. Considering the relatively small patient population it would probably lead to decreased statistical significance if we were to stratify the contribution of different ARDS complication such as VAP or PE

  1. The study limitations should be more described.

The main limitation is the relatively small sample study and being a single center study, both are acknowledged in the manuscript. Other limitations are the lack of data on more advanced ventilation setting such as plateau pressure and driving pressure, the lack of data on vasopressor free days and ventilation free days, again these are also acknowledged. The latter will be addressed on the second part of this study which will be done with data from ARDS patients from external ICU.

Round 2

Reviewer 2 Report

Comments and Suggestions for Authors

The manuscript has been modified. In general, the comments on all-cause mortality and study limitations could be detailed not a response letter but revised manuscript.

Author Response

Many thanks for the comments. My apologies for having submitted and incomplete revision. I should have postponed for health reasons. I hope the revised manuscript is now acceptable. Many thanks

Reviewer 2 

  1. Row 10; word count was lacking.

Added

  1. Row 23; The word ‘CI’ may be proper in ‘95% confidence interval’.

Corrected

  1. Abstract; more keywords can be raised.

Added

  1. Row 47; CXR was abbreviated here. After the sentences, CTX, not chest X-ray, should be always used.

Corrected

  1. Row 77; the characteristics of hospital should be introduced for readers (e.g., bed number, acute medical care-specific, respiratory disease-specialized…). The death rates of C-ARDS could be included in the information.

Added

  1. Row 81; the data resources (e.g., chart-driven or electronic medical recorded-based) could be more demonstrated.

Added.

  1. Row 87; other contributing causes could be more detailed.

Corrected. So any patient with C-ARDS and chest trauma, or pancreatitis, or inhalation injury or community acquired bacterial pneumonia. Any other cause for ARDS apart from C-ARDS was excluded.

  1. Row 89; lack of data…the flow chart (e.g., the number of patients to select in definition) could be necessary. 

Many thanks, we have explained further.

  1. The reader may think whether or not the standardization of skills to obtain the findings by CXR was important between staff.

CXR was segmented to by health care professionals with image experience (an intensive care physician with a background in pulmonology and a radiographer researching deep learning in chest imaging). However, this was completely irrelevant for this study since segmentation of images involved removing all non lung parenquima from chest x rays in order to extract deep learning features. Images that were very rotated, had resuscitation pads or had ECMO cannulas were removed since these decreases he amount of free lung to be assessed and would increase bias. Hence, the short answer is, no, it was not necessary to have particular training to obtain CXR but the second part of this study will involve the RALE score hence it will be necessary to guarantee adequate training and reproducibility of reports. We have acknowledged and changed in line 191 and 104

  1. All-cause mortality; The reader may think what were the types of causes of mortality. Or the cause-specific analyses may produce some findings.

Many thanks it is an interesting point but we have a relatively small sample size. Considering this relatively small patient population it would probably lead to decreased statistical significance if we were to stratify the contribution of different ARDS complication such as VAP or PE

  1. The study limitations should be more described.

We agree. This has been corrected in the last paragraph of discussion

The main limitation is the relatively small sample study and being a single center study, both are acknowledged in the manuscript. Other limitations are the lack of data on more advanced ventilation setting such as plateau pressure and driving pressure, the lack of data on vasopressor free days and ventilation free days, again these are also acknowledged. The latter will be addressed on the second part of this study which will be done with data from ARDS patients from external ICU.